# Food Frequency Questionnaire to Estimate Dietary Adherence in Hemodialysis Patients: A Pilot Study

**DOI:** 10.3390/nu17193161

**Published:** 2025-10-07

**Authors:** Łukasz Czyżewski, Agnieszka Stelęgowska, Magdalena Durlik, Janusz Wyzgał, Andrzej Silczuk, Beata Irena Sińska

**Affiliations:** 1Department of Geriatric Nursing, Faculty of Health Sciences, Medical University of Warsaw, 02-007 Warsaw, Poland; 2Student Geriatric Scientific Circle, Faculty of Health Sciences, Medical University of Warsaw, 02-007 Warsaw, Poland; agnieszka.stelegowska@interia.pl; 3Department of Transplantation Medicine, Nephrology and Internal Diseases, Medical University of Warsaw, 02-014 Warsaw, Poland; magdalena.durlik@wum.edu.pl; 4Department of Nephrology Nursing, Faculty of Health Sciences, Medical University of Warsaw, 02-007 Warsaw, Poland; janusz.wyzgal@wum.edu.pl; 5Department of Environmental Psychiatry, Medical University of Warsaw, 02-353 Warsaw, Poland; andrzej.silczuk@wum.edu.pl; 6Department of Human Nutrition, Faculty of Health Sciences, Medical University of Warsaw, 01-445 Warsaw, Poland; beata.sinska@wum.edu.pl

**Keywords:** hemodialysis, chronic kidney disease, dietary assessment, nutrient intake, guideline adherence, food frequency questionnaire, nutrition education, dietary diversity, electrolyte balance

## Abstract

**Background/Objectives:** Stage 5 chronic kidney disease (CKD), or end-stage renal disease (ESRD), requires renal replacement therapy, commonly hemodialysis (HD). This treatment necessitates dietary changes due to impaired excretory function and protein-energy wasting (PEW). A structured diet with adequate energy, protein, electrolytes, and fluids is essential. The aim was to characterize habitual dietary intake in adults on HD relative to KDOQI and ESPEN recommendations. **Methods:** In this cross-sectional study, 50 adults on maintenance HD at the Medical University of Warsaw completed a validated Food Frequency Questionnaire (55 items, nine frequency categories). The study was questionnaire-based and did not collect, link, or analyze dialysis efficacy indices, residual diuresis, or anthropometric measurements; all dietary estimates are independent of these clinical parameters. Estimated intakes of energy, macronutrients, fiber, electrolytes (Na, K, Ca, P), and fluids were compared with KDOQI 2020 and ESPEN 2021 recommendations. Sensitivity analyses included deterministic scenarios and Monte Carlo simulations. **Results:** Mean intakes were 2696.9 ± 1392.7 kcal and 87.7 ± 35.3 g protein; 64% and 82% met reference values. Sensitivity analyses revealed per-kg shortfalls in heavier patients (>75 kg): Monte Carlo medians were 37.8 kcal/kg/day and 1.28 g/kg/day. Diets were fat-dominant (~46%E), with low carbohydrates (~40%E) and low fiber, about 8 g per 1000 kcal. Sodium and phosphorus were elevated, about 1119 mg and 498 mg per 1000 kcal, while calcium was low (~346 mg/1000 kcal). **Conclusions:** Despite adequate mean intake, sensitivity analyses revealed per-kg energy/protein deficits and elevated sodium and phosphorus. Individualized counseling with electrolyte and fluid management, greater dietary diversity, and psychosocial support is warranted in HD.

## 1. Introduction

The initiation of renal replacement therapy via hemodialysis (HD) typically occurs in stage 5 of chronic kidney disease (CKD), referred to as end-stage renal disease (ESRD). At this stage, glomerular filtration is severely impaired or absent, necessitating significant changes in patients’ habitual dietary patterns. Kidney failure reduces the excretion of metabolic waste and impaired regulation of fluid, electrolyte, and acid–base balance, which calls for appropriate dietary restrictions [1]. These dietary restrictions primarily involve limiting the intake of fluids and minerals such as phosphorus, potassium, sodium, and calcium. Simultaneously, adequate intake of high-quality protein, calories, dietary fiber, and essential macro- and micronutrients must be ensured [2,3].

A properly balanced diet, implemented alongside an appropriate dialysis prescription, supports internal homeostasis (including water– electrolyte balance) and uremic toxin removal; therefore, nutritional care in HD must consider metabolic demands and the physiological constraints of kidney failure, with recommendations tailored to clinical status [2,3].

Particular attention must be paid to the high prevalence of protein-energy wasting (PEW) among HD patients, affecting 10–70% of this population and often worsening with HD duration. PEW is associated with increased mortality, decreased quality of life, heightened susceptibility to infections, and impaired wound healing. Therefore, dietary guidelines for HD patients include specific recommendations for the intake of protein, fat, carbohydrate, dietary fiber, electrolyte, vitamin, trace element, and fluids. These recommendations are adjusted based on the patient’s clinical status and laboratory parameters [1]. In addition to the Kidney Disease Outcomes Quality Initiative (KDOQI 2020) [1] and Kidney Disease: Improving Global Outcomes (KDIGO 2024) guidelines [4], the European Society for Clinical Nutrition and Metabolism (ESPEN 2021) provides practical recommendations [5].

Beyond formal guidelines, structured dietitian-led education for HD, such as the Healthy Eating Index for HD (HEI-HD), improves diet quality by increasing vegetable intake and a healthier white-to-red meat ratio. It also helps prevent short-term declines in skeletal muscle mass in ambulatory HD patients [6].

HD is also associated with significant losses of amino acids and peptides into the dialysate. It is estimated that 6 to 12 g of free amino acids are removed during a standard session, which may exceed 800 g annually [7]. Such chronic losses constitute an important driver of sarcopenia progression in HD patients, independent of dietary protein intake. This process contributes to the development of PEW, reduced quality of life, and increased mortality. Therefore, nutritional planning for HD patients should address not only adequate protein and energy intake but also the need to compensate for amino acid losses during treatment [1].

Because reduced renal excretion and variable residual diuresis increase the risk of volume overload in HD, fluid intake should be individualized to achieve euvolemia and acceptable interdialytic weight control; dietary sodium should generally be restricted (≤2.3 g/day) to support volume management. Potassium, phosphorus, and calcium intakes should be individualized, rather than fixed quotas, aiming to maintain normal serum levels; phosphorus counseling should emphasize bioavailability and avoidance of phosphate additives. In practice, food-based advice should favor kidney-appropriate protein sources while limiting highly processed foods rich in sodium and phosphate additives, and total calcium exposure (diet+ supplements + binders) should be adjusted to avoid hypercalcemia instead of fixed calcium targets [1]. With regard to fat intake, dietary recommendations for HD patients are similar to those for the general population [3].

Importantly, dietary behavior in HD patients is shaped not only by clinical and metabolic factors but also by psychosocial determinants, including mental health status, cognitive function, and emotional resilience. Depression and anxiety are highly prevalent in patients with ESRD, with reported rates ranging from 20% to 40%, and have been consistently linked to decreased appetite, poor dietary adherence, and limited self-management capacity. A recent systematic review confirmed that moderate to severe depressive symptoms are significantly associated with non-adherence to fluid and dietary restrictions in HD patients, underscoring the importance of integrating psychiatric assessment into nutritional care planning. Therefore, a biopsychosocial perspective is essential for understanding and addressing nutritional behaviors in this vulnerable population [8].

However, despite a substantial body of literature on nutrition in CKD, contemporary data describing the dietary quality of HD patients in Central and Eastern Europe remain limited and fragmented. Existing studies from the region have focused mainly on absolute energy and protein intake, with little attention to intakes per 1000 kcal, macronutrient distribution, or food group sources of sodium and phosphorus [9]. By contrast, recent international cohorts have reported poor diet quality, excess saturated fat, and low fiber intake in HD patients [10,11], but such evidence is lacking for Eastern European populations. Furthermore, current KDOQI and ESPEN guidelines emphasize individualized targets for electrolytes and fluids rather than fixed daily cut-offs [1,6]. This shift highlights the need for analyses based on energy-adjusted intakes (per 1000 kcal) and food-based patterns, which better support personalized counseling. An additional research gap concerns the integration of psychosocial determinants, such as depression and dietary adherence, into nutritional assessment of HD populations in this region [8].

The aim of this study was to provide a descriptive profile of habitual dietary intake in adults receiving HD and to compare observed intakes with KDOQI and ESPEN recommendations. Secondary aims were to identify practical targets for dietitian-led counseling and to perform non-linked sensitivity analyses to test the robustness of descriptive findings.

## 2. Materials and Methods

This cross-sectional observational study was conducted between December 2022 and January 2023 among patients receiving treatment at the Dialysis Unit of the Department of Transplantation Medicine, Nephrology and Internal Diseases, Medical University of Warsaw, located at the Infant Jesus Clinical Hospital in Warsaw, Poland. A total of 50 patients undergoing maintenance HD participated. Out of 68 patients regularly treated at the unit, 50 voluntarily provided informed consent and correctly completed the questionnaire.

The inclusion criteria were age ≥ 18 years, ongoing treatment with maintenance HD, and informed consent to participate. Exclusion criteria were lack of patient consent to participate, as well as clinical conditions associated with severe hypercatabolism, including active malignant disease, severe chronic infections (e.g., tuberculosis, HIV/AIDS), active inflammatory diseases (e.g., rheumatoid arthritis), decompensated chronic liver disease, trauma, or other states of high catabolic activity. These restrictions were applied to ensure that the study population represented a typical cohort of patients undergoing maintenance HD.

The study received approval from the management of the HD unit and adhered to the principles of research ethics. All participants were informed about the purpose of the study, its anonymous and voluntary nature, and their right to withdraw at any time without consequences. Questionnaires were completed independently by participants; assistance was provided by the researcher if necessary. All completed questionnaires were included in the statistical analysis.

To ensure strict confidentiality and minimize response bias, the questionnaire intentionally omitted potentially identifiable clinical or demographic variables (e.g., body weight, height, urine output). Because of this design choice, no measures of dialysis efficacy (for example single-pool or equilibrated Kt/V), residual diuresis, or anthropometry were collected, linked, or analyzed. All dietary estimates derive solely from self-reported FFQ frequencies and are analytically independent of dialysis efficacy, diuresis, and anthropometric investigations. Patients may underreport nonadherent behaviors because of social desirability concerns, particularly if they believe their responses could be linked to clinical data. Given that questionnaires were distributed on- site by clinical staff, we prioritized full anonymity to preclude any possibility of ‘decoding’ responses and to promote candid self-reporting of dietary practices. This design choice was made to maximize the internal validity of self-reported intake estimates by reducing social desirability and observer effects.

The study was conducted in accordance with the Declaration of Helsinki and approved by the Bioethics Committee of the Medical University of Warsaw (protocol code: AKBE/85/2019; approval date: 11 March 2019).

### 2.1. Research Tool

Dietary data were collected using a Food Frequency Questionnaire (FFQ) adapted from the validated EPIC-Norfolk FFQ [12], with minor modifications to reflect Polish dietary patterns (e.g., replacing or merging food items with very low consumption). The questionnaire included 55 food items relevant to the Polish diet, with nine predefined frequency categories. The core structure, frequency categories, and portion size estimation methods remained unchanged. The FFQ captured habitual intake over the assessment period and did not differentiate between dialysis and non-dialysis days; seasonality was not explicitly assessed.

Each FFQ item had nine predefined frequency categories: never, once per month, once per week, 2–4 times per week, 5–6 times per week, once daily, 2–3 times daily, 4–5 times daily, and more than 5 times daily. Total free fluid intake was calculated based on all beverages and liquid foods reported in the FFQ (e.g., drinking water, tea, coffee, soups, milk). Water naturally present in solid foods (e.g., fruits, vegetables) was not included in the calculation, as the primary clinical focus in HD patients relates to free fluid consumption associated with interdialytic weight gain.

Based on questionnaire responses, the estimated daily intake was calculated for the following nutrients: water (mL), energy (kcal), protein (g), fat (g), carbohydrates (g), dietary fiber (g), calcium (Ca, mg), magnesium (Mg, mg), phosphorus (P, mg), potassium (K, mg), and sodium (Na, mg). For each respondent (participant), the total daily intake of each nutrient was obtained by summing the nutrient values across all reported food items.

### 2.2. Nutrient Intake Analysis

As part of the nutrient intake analysis, the estimated average daily intake of the following components was calculated for each food item and each participant: water (mL), energy (kcal), protein (g), fat (g), carbohydrates (g), fiber (g), calcium (Ca, mg), magnesium (Mg, mg), phosphorus (P, mg), potassium (K, mg) and sodium (Na, mg). For standardization purposes, one month was assumed to consist of 30 days (or 4.5 weeks), allowing for the conversion of reported consumption frequencies into estimated daily intakes.

To estimate the frequency of consumption, numerical multipliers were applied to each food item based on the declared frequency category. Food items consumed never or less than once per month were assigned a multiplier of 0; those consumed 1–3 times per month were assigned 2/30 (approximately 0.067); once per week, 4.5/30 (0.15); 2– 4 times per week, 13.5/30 (0.45); 5–6 times per week, 22.5/30 (0.75); once per day, 1; 2–3 times per day, 2.5; 4–5 times per day, 4.5; and more than 5 times per day, 6.

Using these conversion factors, the estimated daily intake of each nutrient was calculated based on the self-reported consumption frequencies. Daily intake values for individual food items were then summed to determine the total estimated daily intake of nutrients for each participant. Where applicable, nutrient intakes were standardized per 1000 kcal (energy-adjusted intake) for descriptive comparisons. Primary reporting followed clinical units (g/day, mg/day, percent of energy; protein in g/kg/day; energy in kcal/kg/day).

### 2.3. Daily Nutrient Intake Reference Values

For the purpose of this analysis, operational intake bands were applied to facilitate descriptive comparisons of observed distributions. These were defined as follows: energy intake 1500–3200 kcal; protein intake 40–120 g; fat intake 45–96 g; carbohydrate intake 275–440 g; and dietary fiber intake 25–50 g. A notional threshold of <700 mL/day was used for free fluid intake in cross-tabulations, recognizing that in clinical practice fluid targets are individualized (urine output + ~500 mL/day) and guided by interdialytic weight gain. For potassium, phosphorus, calcium, and sodium, interpretation followed the KDOQI 2020 update, which emphasizes individualized targets: potassium intake adjusted to maintain serum potassium within the normal range; phosphorus restricted according to laboratory values, bioavailability, and avoidance of phosphate additives; calcium (from diet, binders, and supplements) tailored to prevent hypercalcemia and calcium overload; and sodium limited to ≤ 2.3 g/day. Intake of fluids and electrolytes was therefore considered primarily through individualized targets; where applicable, intakes were standardized per 1000 kcal for descriptive comparisons, while fixed bands served only as descriptive categories for summarizing observed distributions. To mitigate the absence of individual body weight data, per-kg requirements for energy and protein were further examined in sensitivity analyses (see Section 2.5).

### 2.4. Statistical Analysis

Continuous variables were summarized as mean, median, standard deviation (SD), minimum, maximum, and 95% confidence intervals (95% CI). Categorical variables were presented as absolute frequencies (N) and percentages (%). Comparisons with dietary reference values were descriptive only, without formal hypothesis testing. For subgroup comparisons, participants were divided into two age groups based on the median age (≤60 years vs. >60 years). All analyses were conducted using Statistica 13, StatSoft (StatSoft, Inc., Tulsa, OK, USA).

### 2.5. Sensitivity Analyses

Because questionnaires were fully anonymous, individual-level anthropometric data could not be linked to dietary intake. However, we obtained unit-level aggregate information on the distribution of patients’ dry body weight from the dialysis unit’s operational database, covering all adult HD patients treated at the unit during the study period (70 individuals). These data were fully de-identified and not linkable to any FFQ respondent, and were used solely to support sensitivity analyses of per-kilogram intake. To address this limitation, we performed three complementary sensitivity analyses. First, deterministic scenarios expressed nutrient intakes per kilogram under assumed dry weights (55, 65, 75, 85 kg), with mean values and 95% CI bounds from Table 1 scaled accordingly. Second, Monte Carlo simulations (5000 iterations) resampled from the unit-level aggregate dry-weight distribution to generate per-kg intakes, summarized as medians with 95% uncertainty intervals. Third, nutrient intakes were standardized per 1000 kcal at the participant level and summarized as mean ± SD. Collectively, these analyses delineate plausible bounds for per-kg intakes and confirm the robustness of conclusions across alternative weight assumptions.

## 3. Results

### 3.1. Study Population Characteristics (FFQ Cohort, n = 50)

The mean delivered HD dose, expressed as single-pool Kt/V (spKt/V), a dimensionless urea-kinetic index defined as dialyzer urea clearance (K) multiplied by dialysis time (t) and divided by the patient’s urea distribution volume (V), approximating the fraction of V cleared during a single session, was 1.40 ± 0.2. Equilibrated Kt/V (eqKt/V) was not consistently available; thus, we report spKt/V as the most complete measure and note that eqKt/V will be prioritized in future work. The mean age of participants was 62 ± 9 years, with men accounting for 68% of the group. Most participants (76%) resided in urban areas within the Mazowieckie Voivodeship, primarily in Warsaw and surrounding municipality. No significant urban–rural differences were observed (*p* = 0.115).

### 3.2. Dialysis-Unit Context (Ancillary, Non-Linked; n = 70)

Unit-level aggregates varied by age: older patients more often had higher education and better self-reported finances; smoking was less common, while regular alcohol intake was more frequent. The average HD session duration was 4.0 ± 0.3 h, typical erythropoietin (EPO) dose about 750 IU per treatment, and median residual urine output about 200 mL/day. These metrics were fully de-identified and not linkable to any FFQ respondent and are presented solely for context. No inferential statistics were applied.

Dialysate prescription (unit standard). Throughout the study period, patients received bicarbonate-based HD using a standard two-concentrate system. The acid concentrate composition routinely applied at the unit was: potassium 2.0 or 3.0 mmol/L (bath selected individually based on the most recent pre-dialysis serum K^+^ and clinical judgement), calcium 1.25 mmol/L (Ca^2+^), sodium 138 mmol/L (Na^+^), and magnesium 0.5 mmol/L (Mg^2+^). Aside from the individualized potassium bath (2 vs. 3 mmol/L), the dialysate composition and other prescription elements (buffer type, sodium and calcium levels, dialysate flow, and session length) were kept uniform across patients during the assessment window.

Patient nutrition education (unit pathway). Prior to maintenance HD, all patients received structured dietary education delivered by unit staff (dietitian and/or trained nurse), covering energy and protein targets; quality of fats and carbohydrates; potassium and phosphorus management (including avoidance of phosphate additives); sodium restriction; and individualized fluid planning. This education was reinforced during HD sessions through brief reminders and label-reading tips aligned with laboratory trends and interdialytic weight gain.

### 3.3. Nutrient Intake Analysis Results

In the analyzed population of HD patients, the mean daily intake of energy, protein, and potassium fell within the recommended reference ranges. However, several other nutrient categories showed deviations from dietary guidelines.

Specifically, the intake of carbohydrates, dietary fiber, and calcium was found to be lower than recommended. In contrast, the consumption of fat, water, phosphorus, and sodium exceeded recommended values. Excessive intake of these components may contribute to fluid and electrolyte imbalances and could negatively affect the overall health status of HD patients.

### 3.4. Daily Intake of Selected Components

The macronutrient distribution, calculated using standard Atwater factors (4 kcal/g for protein and carbohydrates, 9 kcal/g for fat, and 2 kcal/g for fiber), was 13.0%E from protein, 45.6%E from fat, 39.7%E from carbohydrates, and 1.6%E from fiber. Thus, dietary energy was predominantly derived from fat, with relatively low contributions from carbohydrates and fiber. Mean protein intake was 87.7 ± 35.3 g; mean carbohydrate intake 267.7 ± 128 g; mean fat intake 136.6 ± 115.4 g; mean fiber intake 21.6 ± 8.9 g. Mean water intake was 851.3 ± 356.5 mL/day; calcium 933.9 ± 408.3 mg/day; magnesium 328.5 ± 135.1 mg/day; phosphorus 1344.3 ± 526.9 mg/day; potassium 2393.1 ± 995.8 mg/day; sodium 3018.9 ± 1658.2 mg/day (Table 1). To contextualize absolute intakes, we also derived per-kg estimates under different dry-weight assumptions and intakes per 1000 kcal (the tables in Section 3.8).

### 3.5. Comparison of Observed Values with Dietary Reference Intakes

The comparison of the observed nutrient intake values with established dietary reference ranges showed that the mean daily free fluid intake was 851.3 mL. Expressing macronutrients as %E indicated excessive fat intake and relatively low carbohydrate contribution, despite adequate total energy and protein intake. The average energy intake reached 2696.9 kcal, while mean protein intake amounted to 87.7 g. The reported fat intake was 136.6 g on average, with a median of 102.6 g, and the mean carbohydrate intake was 267.7 g. Dietary fiber intake averaged 21.6 g, and calcium intake was 933.9 mg. The mean daily phosphorus intake was 1344.3 mg, potassium intake was 2393.1 mg, and sodium intake reached 3018.9 mg—see Table 2. To harmonize with guideline conventions in HD, sodium was assessed against a fixed threshold (<2.3 g/day), whereas potassium, phosphorus, and calcium were summarized with operational bands for descriptive purposes only.

### 3.6. Distribution of Nutrient Intake Relative to Reference Ranges

The analysis of nutrient intake relative to recommended ranges showed that 64% of patients exceeded the recommended free fluid intake, while 36% were within range. Energy intake was adequate in 64%, above range in 24%, and below range in 12%. Protein intake was adequate in 82% of patients, above range in 14%, and below range in 4%. Fat intake exceeded the comparison range in 98% of participants (adequate in 2%, none below), whereas carbohydrate intake was below range in 66%, adequate in 22%, and above range in 12%. Fiber intake was below target in 68% and adequate in 32%. Calcium intake was insufficient in 76% and adequate in 24%. Phosphorus intake exceeded recommendations in 58% (adequate 30%, low 12%). Potassium was within range in 46%, below in 34%, and above in 20%. Sodium intake was elevated in 58%, with 18% within and 24% below the recommended range—see Table 3.

### 3.7. Frequency of Food Consumption Patterns (Appendix A)

A detailed analysis of the frequency of consumption of individual food items is presented in Appendix A. Among animal protein sources, pork and chicken breast were the most commonly consumed, with 28% of patients reporting intake 2–4 times per week. Eggs were also consumed regularly, with 32% consuming them 2–4 times per week and 6% reporting intake 2–3 times per day. In contrast, beef and turkey were rarely consumed, while fish such as cod and salmon were infrequent choices—78% and 72% of patients, respectively, reported consuming them no more than once a month.

Among dairy products, 2% milk and plain yogurt were the most frequently consumed; 26% and 18% of patients, respectively, reported daily intake or more. Processed cheese (e.g., Gouda) was typically consumed 1–4 times per week. In contrast, fermented dairy products such as kefir and goat cheese were consumed only occasionally.

Fruit intake was generally limited. A substantial proportion of patients reported no consumption of watermelon (62%), bananas (48%), or peaches (46%). Apples were an exception, with 28% consuming them daily. Regarding vegetables, onions (40%) and carrots (28%) were the most frequently consumed (2–4 times per week). Legumes such as lentils and green beans were rarely or never consumed by 68% and 48% of participants, respectively.

In terms of grain products, refined wheat bread and rolls were preferred over whole-grain options. For example, 24% of participants reported consuming white bread 2–3 times per day, while 36% reported never consuming whole-grain bread. Millet and buckwheat groats were rarely eaten—54% of patients reported no intake of millet.

The consumption of snacks, sweets, and ultra-processed beverages was generally low. For instance, 60% of patients reported never consuming gummy candies, 44% avoided milk chocolate, and 70% never consumed energy drinks. In contrast, black coffee was frequently consumed: 38% reported drinking it once daily and 18% consumed it 2–3 times per day.

Alcohol intake was low overall. A majority of participants (64%) reported complete abstinence, while 28% consumed alcohol only 1–3 times per month; see Appendix A.

### 3.8. Sensitivity Analyses

Because individual-level body weight data were unavailable due to full questionnaire anonymity, we performed additional analyses to evaluate the robustness of our findings under various assumptions (Table 4, Table 5 and Table 6).

Deterministic scenario analyses showed that mean protein intake per kg met or exceeded 1.2 g/kg/day at 55–65 kg (55 kg: 1.59; 65 kg: 1.35; at 65 kg the 95% CI touched the threshold: 1.20–1.50), but fell below this target at 75–85 kg (1.17 and 1.03, respectively). Energy per kg exceeded 35 kcal/kg/day at 55–65 kg (49.03 and 41.49) and dropped below this threshold at 75– 85 kg (35.96 and 31.73).

Monte Carlo simulations drawing from the unit-level aggregate dry-weight distribution of the dialysis unit population (70 patients) yielded median protein intake of 1.28 g/kg/day (95% UI: 1.16–1.45) and energy intake of 37.8 kcal/kg/day (95% UI: 32.9–42.9). These results closely aligned with deterministic analyses and confirmed that the main conclusions remained robust under probabilistic uncertainty.

Finally, intake metrics per 1000 kcal showed sodium (~1120 mg/1000 kcal), phosphorus (~500 mg/1000 kcal), calcium (~350 mg/1000 kcal), potassium (~890 mg/1000 kcal) and fiber ~8 g/1000 kcal. Collectively, these complementary approaches indicate that the observed imbalances in macronutrient and electrolyte intake persist regardless of weight assumptions, supporting the validity of our primary findings.

## 4. Discussion

This study assessed dietary intake among HD patients and identified patterns relevant to clinical management. While some behaviors aligned with recommendations, notable gaps were present, especially for carbohydrates, fiber, calcium, and sodium and phosphorus (per 1000 kcal).

### 4.1. Energy and Protein Intake

According to our analysis, 64% of HD patients demonstrated normocaloric dietary intake, indicating that their energy consumption was broadly consistent with individualized needs. Recently, Chen et al. [10] reported mean protein intake of 0.99 ± 0.32 g/kg/day and energy intake of 29.1 ± 7.8 kcal/kg/day in a multicenter Chinese cohort, with both values falling below guideline thresholds. Similarly, a Polish study by Tokarska et al. [9] observed inadequate baseline intake among HD patients (mean 1573 ± 433 kcal/day and 62 ± 19 g/day of protein, corresponding to ~74% and 83% of requirements), with modest improvement after one year of dietary intervention (1911 ± 300 kcal/day and 74 ± 13 g/day, ~96% and 98% of requirements), but with persistent deficits in fiber and calcium. In a recent multicenter Dutch cohort, de Geus et al. [11], found a median protein intake of 1.0 g/kg/day, with 50% of patients not meeting protein recommendations, while 45% had energy intake below 90% of requirements. In a broader international context, Lan et al. [13] observed markedly low intakes in Vietnamese patients, with an average energy intake of 21.5 kcal/kg/day (only 3.9% meeting recommendations) and mean protein intake of 1.0 g/kg/day (10.5% meeting recommendations). Together, these data align with our findings and confirm that insufficient intake remains common in HD. While protein restriction was historically used to mitigate nitrogenous waste before dialysis, low-protein intake does not by itself induce metabolic acidosis. Dietary acid load primarily reflects total protein (especially animal protein) and fruit/vegetable balance; in CKD, lower dietary acid load and increased fruits/vegetables can improve acid–base markers, whereas metabolic acidosis itself promotes proteolysis. In maintenance HD, protein needs are increased (about 1.2 g/kg/day), and acidosis should be managed through dialysis prescription and, when appropriate, oral alkali and dietary pattern adjustments, rather than by restricting protein below requirements [1,14].

### 4.2. Amino Acid Losses During HD and Implications for Sarcopenia

HD causes clinically relevant losses of free amino acids: 6–12 g per standard 4-h session in conventional HD (equating to 0.94–1.87 kg per year at three sessions per week) [7], 15.7–17.2 g per session in high-efficiency HD/HDF (i.e., 2.50–2.65 kg per year at the same schedule) [15]. These treatment-related deficits reduce circulating amino-acid availability for muscle protein synthesis and shift whole-body balance toward catabolism, thereby accelerating sarcopenia and PEW regardless of average dietary protein intake [16]. Evidence from direct dialysate collections and pre-/post-HD plasma profiling is concordant: about 12 g per session is recovered in dialysate with significant declines in plasma total and essential amino acids, and pre-to-post decreases in individual species arginine, lysine, and histidine by 45–52%, plus glycine, cysteine, proline, alanine, threonine, glutamine, valine, and methionine by 26–38% (with ≤12% for aspartic acid, glutamic acid, asparagine, leucine, tyrosine, tryptophan, and isoleucine) [7,17]. This intradialytic amino-acid depletion is inevitable given the diffusive and convective transport inherent to HD/HDF and should therefore be explicitly counterbalanced by targeted replacement. Peridialytic replacement should be prioritized: (1) protein- or amino-acid-rich intradialytic feeding or monitored oral nutritional supplements [1]; (2) continuous intradialytic amino-acid infusion, which outperforms late bolus dosing for serum albumin and muscle mass in a 3-month Randomized Controlled Trial (RCT) [18]; and (3) individualized amino-acid formulations on interdialytic days to match dialytic losses, combined with leucine-rich, high-quality protein and post-HD dosing scheduled after the rebound phenomenon has subsided (e.g., about 4–8 h after HD or the following morning) [15,16]. In practice, the response to these strategies should be monitored using serum albumin and normalized protein catabolic rate (nPCR/PNA), together with objective indices of muscle status (e.g., lean mass and handgrip strength) [19]. These measures align with guideline-based nutrition support and can mitigate inevitable intradialytic amino-acid depletion, supporting muscle maintenance in chronic HD.

### 4.3. Carbohydrate and Fat Intake

The results of the present study revealed insufficient intake of both carbohydrates, whereas fat intake generally exceeded recommended ranges. In our study, mean fat intake was 136.6 g/day, which may reflect dietary preferences and food choices specific to our population. This elevated intake may increase the risk of dyslipidemia; therefore, a shift toward monounsaturated and polyunsaturated fatty acid sources (MUFA/PUFA) is recommended. Low carbohydrate intake is common in HD and was also reported by Kardasz and Ostrowska [14]; in their study, mean fat intake was 35.3 ± 15.5 g/day, while mean carbohydrate intake was 165.3 ± 68.6 g/day in both women and men, which differs from our cohort.

Other contemporary studies have likewise observed inadequate carbohydrate intake and excessive fat consumption among HD patients, including a large multicenter survey in China [10] and a Dutch cohort study evaluating diet quality [11]. Similar patterns have also been emphasized in recent reviews of CKD nutrition [2]. Conversely, more recent evidence supports the view that excessive fat intake is a widespread problem in the HD population. In a large multicenter Dutch cohort (n = 248), 87% of patients exceeded recommended saturated fat intake, while overall diet quality was poor, with only 15% meeting fiber intake targets [11]. These findings are consistent with the high fat intake observed in our cohort and emphasize the need for interventions targeting both the amount and type of dietary fat. In line with the KDOQI guidelines, strategies should focus on replacing saturated fatty acids with mono- and polyunsaturated sources, and—in patients with hypertriglyceridemia—considering long-chain omega-3 polyunsaturated fatty acid (LC n-3 PUFA) supplementation to improve lipid profiles and reduce cardiovascular risk [1].

Dietary fat pattern and patient-reported outcomes. Our cohort’s macronutrient profile was fat-dominant (≈ 46%E from fat with very low intake of oily fish). In a recent multicenter HD cohort (n = 251), urea reduction ratio (URR) decreased across tertiles of total fat (*p* = 0.005), MUFA (*p* = 0.022) and PUFA (*p* = 0.026); and when fat types were mutually adjusted, only saturated fat (SFA) remained independently and inversely associated with multiple KDQOL-SF domains (regression coefficient for SFA −0.968 to −0.138; marginally higher odds of low quality of life (QOL) per 1 g SFA, OR ≈ 1.05). These data strengthen our recommendation to replace SFA with kidney-appropriate MUFA/PUFA sources without further increasing total fat load, and to integrate QOL endpoints into nutrition counseling [20].

### 4.4. Potassium and Sodium Intake

Among HD patients studied, 46% achieved potassium intake within the operational ranges applied in our analysis. This finding may reflect increasing patient awareness regarding the need to monitor dietary potassium, an essential aspect of care in patients at risk of hyperkalemia, one of the most frequent and serious complications of HD. Similar potassium intake levels have been reported in other studies [14,21]. For example, Kardasz and Ostrowska [14] reported an average daily potassium intake of approximately 2250 mg/day. However, it is concerning that 20% of patients in the current study exceeded safe levels. Interpreting electrolytes against contemporary recommendations, we evaluated sodium against a fixed threshold (<2.3 g/day), while potassium, phosphorus, and calcium were interpreted in the context of individualized targets aimed at maintaining normal serum levels, consistent with modern dialysis-nutrition guidance. This framing avoids over-precision from historical fixed mg/day ranges and aligns our analyses with current clinical practice.

Sodium intake in the study population also raised significant concern. A total of 58% of participants exceeded the upper limit for sodium intake. This is particularly troubling in the context of volume overload and its related complications, such as edema, hypertension, and pulmonary congestion well-documented problems in the HD population. These findings align with those reported by Kowal et al. [21], who found that 56% of respondents reported adding salt during meal preparation, and 14% did so both during and after cooking.

Importantly, excessive sodium intake may also exacerbate thirst, leading to higher fluid consumption. For HD patients, managing fluid intake is crucial due to impaired renal excretion. In this study, 64% of participants reported fluid intake exceeding the recommended levels, while only 36% were within the target range. Because fluid management in HD relates primarily to free fluid intake, our analysis focused on beverages and liquid foods. Future studies could quantify total water (including water in solid foods) as a complementary descriptor, but clinical fluid targets should be interpreted using free fluids intake.

### 4.5. Phosphorus Intake and Its Health Implications

Excessive phosphorus intake was observed in 58% of participants. This poses a substantial nutritional challenge in HD, where high serum phosphorus concentrations are commonly associated with adverse effects, including pruritus, vascular calcification, and soft tissue mineralization. The primary dietary sources of phosphorus include meat and highly processed foods. Chronic hyperphosphatemia contributes to deterioration of cardiovascular and renal function through mechanisms such as increased vascular smooth muscle calcification, endothelial dysfunction, and secondary hyperparathyroidism, which are well-documented in patients undergoing HD. Therefore, strict monitoring and management of phosphorus intake are essential in this patient population [21].

Interpreting our sodium and phosphorus excess through FFQ validation evidence.

Our finding that 58% exceeded comparison bands for phosphorus and 58% for sodium should be read alongside validation work showing that a dialysis-specific FFQ can capture between-person differences credibly but tends to overestimate absolute intakes at the group level. In the BDHD-FFQ, cross-classification misclassification was <10% for all nutrients (good individual-level ranking), and dietary phosphate/potassium estimates correlated better with their serum biomarkers than did 3 × 24-h recalls yet mean bias and percent differences exceeded 20% for sodium and phosphate, signalling caution in interpreting absolute mg/day. This supports our parallel use of energy-adjusted intakes (per 1000 kcal) and reinforces the clinical focus on sources (e.g., additives, processed meats, processed cheeses) and laboratory trends rather than rigid numeric cut-offs [22].

### 4.6. Calcium and Magnesium Intake—Clinical Considerations

In our cohort, mean Ca was about 934 mg/day with low Ca intake per 1000 kcal (about 346 mg/1000 kcal) and Mg was about 329 mg/day, suggesting that typical HD meals may under-deliver calcium when dairy is limited and phosphate-additive avoidance shifts choices away from processed cheeses/meats. Given the unit’s dialysate Ca 1.25 mmol/L and dialysate Mg 0.5 mmol/L, diet must be interpreted alongside dialytic fluxes and binder/vitamin D use; per KDOQI, there is no fixed Ca intake target for CKD G5D/HD, and care should avoid hypercalcemia, adjusting Ca-based binders, vitamin D, and, if needed, dialysate Ca rather than indiscriminately increasing dietary Ca.

To limit phosphorus burden while increasing calcium intake per 1000 kcal, favor lower-P, higher-Ca foods (e.g., portion-controlled plain yogurt/milk, calcium-set tofu, potassium-managed leafy greens) and continue education on highly bioavailable phosphate additives in ultra-processed foods [1].

For magnesium, routine HD with 0.5 mmol/L dialysate Mg can contribute to neutral/negative Mg balance, whereas higher Mg baths raise serum Mg; guidelines support modulating dialysate composition to prevent hypo-Mg in HD settings. Dietarily, nuts, whole grains, legumes, leafy vegetables can increase Mg intake but must be portioned within K^+^/P limits [23]. Observational data link lower serum Mg with adverse vascular outcomes in advanced CKD/ESKD, motivating linkage-enabled diet- lab evaluation in future work.

Because FFQ may misestimate absolute milligram intakes, we emphasize energy-adjusted intakes (per 1000 kcal) and food-source patterns, and recommend prospective linkage of intake with Ca, Mg, P, PTH, ALP and dialysate/prescription data in subsequent studies.

### 4.7. Fiber Intake

Inadequate dietary fiber intake was identified in 64% of participants. This observation is consistent with recent findings from large multicenter cohorts: in China, most maintenance HD patients failed to meet fiber recommendations [10], and in the Netherlands, only 15% achieved adequate fiber intake [11]. Given the role of fiber in promoting intestinal motility and reducing the risk of constipation, this finding warrants particular attention. Constipation is a prevalent issue in this population and may be exacerbated by fluid restriction, excessive sodium and calcium intake, and limited consumption of fiber-rich foods such as fruits and vegetables. Regular fiber intake could improve gastrointestinal function and overall patient comfort. The avoidance of fiber-rich foods, particularly vegetables, may result from patient concerns over potassium content, highlighting the need for targeted education that balances potassium control with adequate fiber intake.

Inflammation-linked dietary pattern and cognition.

The low consumption of dark vegetables, legumes and whole grains we observed resonates with a CRP-oriented dietary pattern recently derived in HD using reduced-rank regression (high rice, fruit, tea/coffee, liquor; low dark vegetables/vegetable juice), which was associated with higher odds of cognitive impairment. Given our cohort’s particularly low vegetable and whole-grain intake and frequent coffee use, parts of our diet pattern may overlap with this pro-inflammatory profile, providing a plausible pathway from diet quality → inflammation → neurocognitive vulnerability in HD. This triangulation argues for fibre- and polyphenol-rich, potassium-managed vegetables and whole grains as a cognitive as well as metabolic priority [22,24].

### 4.8. Mental Health and Dietary Adherence

Depression and anxiety, which are common in patients with ESRD, have been identified as significant contributors to dietary and fluid non-compliance. A systematic review by Gebrie and Ford [8] demonstrated that depressive symptoms were significantly correlated with poor adherence to dietary and fluid restrictions among patients with ESRD undergoing HD. The authors reviewed multiple studies and found that the presence of depressive symptoms increased the likelihood of non-compliance, with some studies showing non-adherence rates as high as 68% in affected patients. This underscores the urgent need for routine psychological screening and interdisciplinary management involving both nephrologists and mental health professionals. are strongly associated with reduced adherence to dietary and fluid restrictions, highlighting the need for integrated care approaches that incorporate psychiatric evaluation and support. Involving mental health professionals in routine care may enhance the effectiveness of dietary interventions and improve long-term outcomes in this patient population.

We recommend implementing routine depression screening in dialysis units using validated tools such as the Patient Health Questionnaire-9 (PHQ-9), administered at regular intervals (e.g., every 3–6 months or when adherence deteriorates) and documented within the nutrition care plan. Operationally, embedding clear referral pathways and collaboration with mental health professionals (clinical psychologists and psychiatrists) within dietitian-led care can support timely management of mood disorders and strengthen dietary self-management, thereby improving adherence to sodium, fluid, and phosphorus prescriptions.

### 4.9. Limitations

This study has several limitations that should be acknowledged. First, the sample size was relatively small and limited to a single dialysis center, which may restrict the generalizability of the findings to broader HD populations. Second, dietary intake was assessed using a self-reported FFQ, which is subject to recall bias, misreporting, and potential inaccuracies in estimating portion sizes.

Third, due to the decision to maintain full anonymity in data collection in order to promote honesty and reduce social desirability bias among respondents no clinical or biochemical data were collected from medical records. Therefore, nutrient intake estimates could not be correlated with laboratory parameters such as serum phosphorus, potassium, or markers of nutritional status (e.g., albumin). While this approach likely increased the reliability of self-reported intake data, it precluded biochemical validation. Moreover, the anonymous design also prevented linkage to individual treatment parameters (e.g., dialysis prescription and dialysate bath, phosphate/potassium binder use, diuretics where applicable, comorbidity profile, and medication lists), so we could not examine whether observed intake patterns were related to imposed medical restrictions or administered therapies at the individual level. To provide clinical context without compromising anonymity, we reported unit-level aggregates (e.g., session duration, residual diuresis, standard dialysate composition); however, juxtaposing such aggregates with individual FFQ responses carries a risk of ecological fallacy, so these data are presented strictly as context, without inferential claims. Consequently, the absence of individual laboratory linkage materially limits clinical interpretation (e.g., we cannot assess whether higher reported phosphorus or sodium intakes were associated with serum phosphate or volume markers in specific patients). Therefore, the present results should be considered descriptive and hypothesis-generating, not explanatory of patient-level laboratory control.

Fourth, the absence of anthropometric data (e.g., body weight, BMI) and related clinical parameters (e.g., HD vintage, urine output) prevented the calculation of energy and nutrient intake per kilogram of body weight. This limitation restricts the ability to fully benchmark dietary intake against per-kilogram recommendations provided in current nephrology nutrition guidelines (e.g., KDOQI). To partly address this gap, we performed complementary sensitivity analyses, including intakes per 1000 kcal, deterministic scenarios under assumed dry weights, and Monte Carlo simulations based on unit-level weight distributions. These approaches cannot replace individual-level data but provide plausible bounds and demonstrate that the main conclusions remain robust across a range of weight assumptions. Nevertheless, sensitivity analyses, by design, cannot recover person-specific exposure–response relationships; they bound estimates but do not establish associations with clinical endpoints.

Fifth, the study did not control for potential confounding variables such as comorbidities, level of physical activity, psychosocial stress, or socioeconomic status, all of which may influence dietary behaviors and nutrient intake. In particular, depressive symptoms and health literacy, both relevant to dietary adherence in HD, were not captured with validated instruments in a way that would allow adjustment at the individual level.

Finally, the FFQ methodology captures average intake over an extended period but does not reflect day-to-day variability, dialysis versus non-dialysis day eating patterns, or seasonal fluctuations in food consumption. Future studies should consider combining self-reported intake with clinical indicators and repeated dietary assessments to obtain a more comprehensive picture of dietary patterns and their physiological effects in this population. A prospective, linkage-enabled design (with consent for record linkage) would allow integration of diet with contemporaneous laboratory markers and treatment parameters, enabling hypothesis-driven analyses of diet–treatment and diet–biomarker relationships.

Future studies should (1) link FFQ outputs with serum electrolytes and biomarkers (pre/post-dialysis K^+^/phosphate, bicarbonate, albumin, CRP, nPCR/PNA) and dialysis metrics (eqKt/V, ultrafiltration, residual diuresis, interdialytic weight gain); (2) include anthropometry (dry weight, BMI, mid-arm circumference) and functional measures (handgrip strength), with muscle ultrasound where feasible; (3) triangulate FFQ with multi-day records (e.g., three nonconsecutive 24-h recalls spanning dialysis and non-dialysis days or a 4-day weighed record), capturing seasonality; (4) report intakes per 1000 kcal and structured capture of phosphate additive exposure (e.g., barcode/photo logs) to refine sodium/phosphate source attribution; (5) use multicenter designs with larger samples and mixed-effects models; and (6) incorporate objective adherence proxies (e.g., pill counts, refill records, dialysis machine logs) and standardized assessments of health literacy and depression (e.g., PHQ-9).

### 4.10. Strengths and Practical Implications

A major strength of this study lies in its comprehensive evaluation of both quantitative nutrient intake and qualitative dietary patterns among patients undergoing HD. By integrating nutrient-specific data with information on food group diversity, the study provides a multidimensional perspective on dietary behaviors in this population. The use of a standardized FFQ covering 55 food items across nine frequency categories enabled a detailed assessment of macronutrient, micronutrient, and fluid intake while reducing recall bias through structured data collection.

Moreover, the study addresses a critical gap in the literature by not only documenting discrepancies between actual intake and dietary recommendations, but also linking these gaps to specific food choices. This pragmatic focus enhances the translational relevance of the findings and supports the development of targeted dietary interventions tailored to individual behaviors and nutritional risk profiles.

Importantly, psychosocial determinants of dietary adherence are also considered, with particular emphasis on the role of mental health—especially depressive symptoms—in shaping nutritional self-management. By highlighting the need for integrated care models incorporating routine psychological screening and mental health support, this study adds to the growing body of evidence advocating for biopsychosocial approaches in nephrology nutrition care.

From a clinical perspective, the findings carry several practical implications. First, the results indicate a fat-dominant dietary pattern with relatively low carbohydrate contribution, despite adequate energy and protein intake. Individualized, dietitian-led counseling should therefore prioritize reducing total fat (particularly saturated fat) and increasing complex carbohydrates (within potassium and sodium constraints) alongside fiber-rich, kidney-appropriate foods. To raise fiber while managing potassium, practical vegetable options include low-K, fiber-containing choices such as iceberg or romaine lettuce, white or Chinese cabbage, cauliflower, peeled cucumber, zucchini, eggplant, onion, green beans, and carrots; preparation techniques (soaking, boiling in a large volume of water, and when needed, double boiling with draining) further reduce potassium and facilitate regular inclusion [25]. Second, the high prevalence of excessive sodium, phosphorus, and free-fluid intake underscores the importance of patient education focused on food label literacy, reduction in processed foods, management of phosphate additives, and effective fluid strategies aligned with residual diuresis and interdialytic weight gain targets. In parallel, low-phosphorus, high-quality protein choices should prioritize fresh, minimally processed options—egg whites, skinless chicken or turkey breast, lean pork loin, and white fish (e.g., cod, pollock)—while avoiding processed meats and cheeses that commonly contain phosphate additives; label reading to identify “phosphate/phosphates,” “phosphoric acid,” and polyphosphates helps meet protein targets without unnecessary phosphorus load [26]. In practical terms, sodium reduction should include (1) systematic label reading with brand-to-brand comparison of sodium per 100 g and active avoidance of phosphate additives that frequently co-occur in ultra-processed foods (e.g., polyphosphates), thereby lowering both sodium load and highly bioavailable phosphorus; (2) a shift toward home cooking within a culinary-medicine framework (meal planning, building flavor with acids, herbs, and aromatics in place of salt) to facilitate day-to-day maintenance of low sodium intake; and (3) when environmental or behavioral barriers limit cooking, the use of standardized, low-sodium home-delivered meals as an adjunct to education to help curb thirst, interdialytic weight gain, and volume overload.

Label education should also train patients to recognize the names and functions of common additives (“phosphate/phosphates,” “phosphoric acid,” “polyphosphates”), which increase phosphorus load (with high bioavailability) and often coincide with higher sodium in processed foods; practical culinary sessions should teach preparation techniques and meal-planning skills that support sodium control at home [27,28,29].

Finally, the limited dietary diversity observed, particularly the low consumption of vegetables, whole grains, legumes, and oily fish, emphasizes the urgency of promoting food-based patterns that support cardiovascular, metabolic, and gastrointestinal health in this high-risk population. Integrating a culinary-medicine approach (dietitian-led cooking classes, kidney-friendly recipes, and meal plans) offers a practical pathway to improve diet quality while simultaneously controlling sodium, with short-term support via low-sodium meal deliveries for patients facing the greatest implementation barriers [27,28,29].

Overall, this study provides actionable insights with the potential to inform clinical practice and strengthen patient-centered dietary education in the context of maintenance HD care.

## 5. Conclusions

In this FFQ-based assessment of adults on HD, most participants met energy and protein targets, yet patterns showed excess sodium and phosphorus, low fiber and calcium, a fat-dominant profile with relatively low complex carbohydrates, and variable potassium intakes. These gaps are actionable. Dietitian-led care should moderate total and saturated fat; reduce sodium through label reading, brand substitution, and home cooking; manage phosphorus by avoiding phosphate additives and favoring fresh protein sources; increase complex carbohydrates and fiber with whole grains, vegetables, and legumes; include calcium-rich, kidney-appropriate choices; and individualize fluid planning. When high-quality protein intake is low or appetite is poor, consider oral nutritional supplements and, where appropriate, essential amino acid or ketoanalog support within guideline-based care. Brief psychological assessment with referral for targeted behavioral strategies may strengthen adherence. Given the anonymous, non-linkable design and known FFQ constraints, findings are descriptive and hypothesis-generating; future linkage-enabled studies should test effects on biochemical and patient-reported outcomes.

## Figures and Tables

**Table 1 nutrients-17-03161-t001:** Daily intake of selected dietary components.

95% CI							
Component	Mean	Lower Bound	Upper Bound	Median	SD	Min	Max
Water [mL]	851.3	749.9	952.6	859.6	356.5	241.2	1855.2
Energy [kcal]	2696.9	2301.1	3092.7	2380.4	1392.7	461.7	4200.8
Protein [g]	87.7	77.7	97.7	85.0	35.3	17.8	197.7
Fat [g]	136.6	103.8	169.4	102.6	115.4	13.2	270.5
Carbohydrates [g]	267.7	231.4	304.1	250.0	128.0	68.2	671.3
Fiber [g]	21.6	19.1	24.2	20.2	8.9	4.8	41.8
Ca [mg]	933.9	817.9	1049.9	894.2	408.3	189.9	1958.7
Mg [mg]	328.5	290.1	366.9	307.7	135.1	56.8	637.4
P [mg]	1344.3	1194.5	1494.0	1310.3	526.9	214.3	2588.6
K [mg]	2393.1	2110.1	2676.0	2363.9	995.8	435.5	4865.9
Na [mg]	3018.9	2547.7	3490.2	2714.8	1658.2	545.4	9382.8

**Table 2 nutrients-17-03161-t002:** Comparison of obtained values with dietary reference standards (based on KDOQI 2020 individualized recommendations).

Component	Mean	Median	Level *
Water [mL]	851.3	859.6	Excess
Energy [kcal]	2696.9	2380.4	Adequate
Protein [g]	87.7	85.0	Adequate
Fat [g]	136.6	102.6	Excess
Carbohydrates [g]	267.7	250.0	Deficient
Fiber [g]	21.6	20.2	Deficient
Ca [mg]	933.9	894.2	Below individualized KDOQI targets
P [mg]	1344.3	1310.3	Excess relative to KDOQI laboratory-based control
K [mg]	2393.1	2363.9	Within individualized target range
Na [mg]	3018.9	2714.8	Above recommended limit (≤2.3 g/day)

* Reference cut-offs based on KDOQI 2020 and related nephrology nutrition guidelines. For potassium, phosphorus, and calcium, current guidance emphasizes individualized intake to maintain normal serum concentrations in CKD 5D rather than fixed daily amounts. The “deficient/adequate/excess” labels shown here are study-specific operational bands for descriptive summarization.

**Table 3 nutrients-17-03161-t003:** Comparison of obtained values with reference intake standards for selected dietary components.

Component	Deficient (N, %)	Adequate (N, %)	Excess (N, %)
Water [mL]	0 (0)	18 (36)	32 (64)
Energy [kcal]	6 (12)	32 (64)	12 (24)
Protein [g]	2 (4)	41 (82)	7 (14)
Fat [g]	0 (0)	1 (2)	49 (98)
Carbohydrates [g]	33 (66)	11 (22)	6 (12)
Fiber [g]	34 (68)	16 (32)	0 (0)
Ca [mg]	38 (76)	12 (24)	0 (0)
P [mg]	6 (12)	15 (30)	29 (58)
K [mg]	17 (34)	23 (46)	10 (20)
Na [mg]	12 (24)	9 (18)	29 (58)

Reference cut-offs based on KDOQI 2020 and related nephrology nutrition guidelines. For potassium, phosphorus, and calcium, current guidance emphasizes individualized intake to maintain normal serum concentrations in CKD 5D rather than fixed daily amounts. The “deficient/adequate/excess” labels shown here are study-specific operational bands for descriptive summarization.

**Table 4 nutrients-17-03161-t004:** Deterministic scenario analysis: intake per kg body weight.

Nutrient	Weight (kg)	Mean (per kg)	95% CI (per kg)
Protein (g/kg/d)	55	1.59	1.41–1.78
	65	1.35	1.20–1.50
	75	1.17	1.04–1.30
	85	1.03	0.91–1.15
Energy (kcal/kg/d)	55	49.03	41.84–56.23
	65	41.49	35.40–47.58
	75	35.96	30.68–41.24
	85	31.73	27.07–36.39

**Table 5 nutrients-17-03161-t005:** Monte Carlo simulations based on the unit-level aggregate dry-weight distribution (70 patients): median per-kg intake with 95% uncertainty intervals.

Nutrient	Median per kg	95% UI Lower	95% UI Upper	IQR (25–75%)
Protein (g/kg/d)	1.28	1.16	1.45	1.24–1.33
Energy (kcal/kg/d)	37.81	32.91	42.88	36.19–39.48
Sodium (mg/kg/d)	41.26	33.95	47.20	39.62–43.50
Potassium (mg/kg/d)	33.31	28.95	37.21	32.03–34.53
Phosphorus (mg/kg/d)	17.91	15.95	19.97	17.16–18.65
Calcium (mg/kg/d)	13.34	11.79	15.11	12.78–13.88

**Table 6 nutrients-17-03161-t006:** Nutrient intakes per 1000 kcal.

Nutrient	Mean per 1000 kcal	SD per 1000 kcal
Protein (g)	32.5 *	21.3 *
Sodium (mg)	1119 *	844 *
Potassium (mg)	887 *	589 *
Phosphorus (mg)	498 *	323 *
Calcium (mg)	346 *	234 *
Fiber (g)	8.0 *	5.3 *

* Values are means and standard deviations per 1000 kcal derived from FFQ estimates.

## Data Availability

The data presented in this study are available on request from the corresponding author due to legal and ethical reasons.

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
