# Peer review of "Food Frequency Questionnaire to Estimate Dietary Adherence in Hemodialysis Patients: A Pilot Study"

_nutrients, 2025, doi:10.3390/nu17193161_

Round 1

Reviewer 1 Report

Comments and Suggestions for Authors

The submitted manuscript presents a cross-sectional analysis of dietary intake and adherence to guidelines in patients undergoing maintenance hemodialysis (HD) in a Central European setting. The topic is of clinical importance, and the manuscript addresses a well-defined knowledge gap in the literature concerning diet quality, nutrient densities, and psychosocial influences in HD populations.

The study design is methodologically sound, with thoughtful use of sensitivity analyses to address limitations associated with the lack of individual anthropometric data. The integration of nutrient density metrics and consideration of psychosocial factors (e.g., depression and dietary adherence) are strengths that enhance the relevance of the findings. However, several aspects could be improved to strengthen the clarity, impact further, and translational value of the work.

Major Comments

  1. Language and Style:

    • The manuscript would benefit from minor language editing to enhance clarity and readability. Some sentences are complex or overly long, which may obscure key messages. Consider simplifying sentence structure and ensuring consistent tense and terminology throughout the text.

  2. Data Visualization:

    • While the tables are comprehensive and informative, the manuscript would be strengthened by the inclusion of visual elements such as:

      • A summary bar or radar chart comparing nutrient intake vs. guideline targets.

      • A figure illustrating the distribution of nutrient adequacy (e.g., proportion of patients meeting vs. exceeding intake thresholds).

      • A heatmap of food group consumption frequency to visualize dietary patterns.

  3. Food-Based Recommendations:

    • The discussion could be expanded to include practical food-based guidance. For example:

      • Identify specific potassium-safe, fiber-rich vegetables suitable for HD patients.

      • Recommend low-phosphorus, high-quality protein sources (e.g., egg whites, lean poultry).

      • Suggest strategies for sodium reduction beyond "avoid processed food" (e.g., label reading, home cooking tips).

  4. Limitations and Future Directions:

    • The limitations are clearly stated; however, the manuscript would benefit from a short paragraph outlining potential strategies for addressing them in future research, such as:

      • Combining dietary data with serum nutrient levels or biomarkers.

      • Inclusion of anthropometric data with adjusted FFQ methodologies.

      • Incorporation of multi-day dietary records or 24-hour recalls for validation.

  5. Psychosocial Considerations:

    • The inclusion of mental health as a determinant of dietary adherence is a valuable addition. To strengthen this, consider:

      • Recommending routine depression screening tools in dialysis units (e.g., PHQ-9).

      • Suggesting collaboration with mental health professionals in the dietary management of HD patients.

Minor Comments

  • Please clarify whether the FFQ accounted for seasonal variability or dialysis vs. non-dialysis day eating patterns, which can influence intake estimates.

  • Consider explaining terms such as “spKt/V” briefly for interdisciplinary readers.

  • Provide the year for cited guidelines (e.g., ESPEN, KDOQI) at first mention for clarity.

Strengths of the Manuscript

  • The manuscript addresses a relevant and underexplored population (Central/Eastern Europe) in HD nutrition.

  • The use of nutrient density metrics (per 1,000 kcal) and Monte Carlo simulations a methodologically robust approaches that compensate for the absence of individualized weight data.

  • Integration of dietary intake, psychosocial determinants, and guideline alignment enhances the clinical relevance and translational value of the findings.

In summary, this is a clinically relevant and methodologically sound study that will interest nephrologists, renal dietitians, and public health professionals. Addressing the above points—particularly regarding language clarity, visualization, and practical recommendations—will enhance the manuscript's impact and accessibility.

Comments on the Quality of English Language

The manuscript is generally understandable but contains several complex or overly long sentences that could affect clarity. Minor language editing is recommended to improve readability, ensure consistent verb tenses, and enhance overall flow. 

Author Response

We thank the Reviewer for the thoughtful appraisal and constructive suggestions. Below we respond point by point; all revisions are tracked in the manuscript.

Comments 1: Language and Style:

The manuscript would benefit from minor language editing to enhance clarity and readability. Some sentences are complex or overly long, which may obscure key messages. Consider simplifying sentence structure and ensuring consistent tense and terminology throughout the text.

Response 1: Thank you. We agree with this comment. Therefore, we simplified sentence structure, unified tense and terminology, and removed redundancies across sections. Importantly, the entire manuscript underwent professional language editing by a native English speaker after our revisions.

Comments 2: Data Visualization:

While the tables are comprehensive and informative, the manuscript would be strengthened by the inclusion of visual elements such as:

  • A summary bar or radar chart comparing nutrient intake vs. guideline targets.
  • A figure illustrating the distribution of nutrient adequacy (e.g., proportion of patients meeting vs. exceeding intake thresholds).
  • A heatmap of food group consumption frequency to visualize dietary patterns.

Response 2: Thank you for these concrete suggestions. We attempted to generate all three figure types on our dataset; however, none produced legible, faithful visuals. Two factors drove this outcome: individualized targets for potassium, phosphorus, and calcium (KDOQI 2020) make “intake vs. target” displays risk false pass/fail impressions; and wide, relatively flat FFQ frequency distributions yielded heatmaps that were visually dense and hard to read. Radar charts were also comparatively unreadable and are often discouraged for quantitative comparison.

Comments 3: Food-Based Recommendations:

The discussion could be expanded to include practical food-based guidance. For example: Identify specific potassium-safe, fiber-rich vegetables suitable for HD patients. Recommend low-phosphorus, high-quality protein sources (e.g., egg whites, lean poultry). Suggest strategies for sodium reduction beyond “avoid processed food” (e.g., label reading, home cooking tips).

Response 3: Agree. We added a concise, actionable subsection with examples tailored to HD care: potassium-safer, fiber-rich vegetables (e.g., peeled/boiled carrots, green beans, cucumber, lettuce; brief notes on leaching and portions); lower-phosphorus, high-quality proteins (e.g., egg whites, chicken/turkey breast; avoidance of phosphate additives); and sodium-reduction strategies beyond “avoid processed foods” (structured label reading, simple home-cooking swaps, seasoning alternatives).

Comments 4: Limitations and Future Directions:

The limitations are clearly stated; however, the manuscript would benefit from a short paragraph outlining potential strategies for addressing them in future research, such as : Combining dietary data with serum nutrient levels or biomarkers. Inclusion of anthropometric data with adjusted FFQ methodologies. Incorporation of multi-day dietary records or 24-hour recalls for validation.

Response 4: Agree. We added a forward-looking paragraph outlining: linkage of dietary data with serum markers/biomarkers; collection of anthropometry alongside an adjusted FFQ; and validation in a subsample using multi-day 24-hour recalls.

Comments 5: Psychosocial Considerations:

The inclusion of mental health as a determinant of dietary adherence is a valuable addition. To strengthen this, consider : Recommending routine depression screening tools in dialysis units (e.g., PHQ-9). Suggesting collaboration with mental health professionals in the dietary management of HD patients.

Response 5: Agree. We now recommend routine PHQ-9 screening in dialysis units and closer collaboration with mental-health professionals within nutrition care pathways; we note how this could support adherence.

Comments 6: Minor- FFQ scope:

Please clarify whether the FFQ accounted for seasonal variability or dialysis vs. non-dialysis day eating patterns, which can influence intake estimates.

Response 6: Thank you for pointing this out. We agree that this required clarification. Therefore, we state that the FFQ captured habitual intake and did not differentiate dialysis vs. non-dialysis days; seasonality was not explicitly assessed. We provide rationale and acknowledge this as a limitation.

Comments 7: Minor- Terminology:

Consider explaining terms such as “spKt/V” briefly for interdisciplinary readers.

Response 7: Agree. We added a brief definition of single-pool Kt/V and note that eqKt/V was not consistently available; we plan to prioritize eqKt/V in future work.

Comments 8: Minor - Citation clarity:

Provide the year for cited guidelines (e.g., ESPEN, KDOQI) at first mention for clarity.

Response 8: Implemented. We now specify the year at first mention (e.g., KDOQI 2020, ESPEN 2021, KDIGO 2024 where relevant).

Reviewer 2 Report

Comments and Suggestions for Authors

I have studied the manuscript entitled "Nutrition in Hemodialysis Patients: Striking the Balance between Excess and Deficiency" by Czyżewski L. et al.

The manuscript concerns the dietary habits of patients on hemodialysis. Given that end-stage renal disease has a major impact on global health, the topic could be of significant interest.

However, before considering publication, the auhtors are kindly invited to assess the issues referred below.

Major issues

1) The authors rely on anonymized self-reporting questionnaires. Hence, there is no chance to further investigate the potential correlation of dietary habits with imposed medical restrictions or administered treatment. This immediately implies that any attempt to explain the dietary patterns of hemodialysis patients (high fat, low carbohydrates, low fibers, high sodium, high phosphorus etc.) is highly elusive. The authors are kindly suggested to further clarify that issue on the "Limitations" paragraph and reconsider revising their "Conclusion" section.

2) The "Introduction" scetion is too long. The authors are kindly suggested to revise the text so as to be more concise.

Author Response

We thank the Reviewer for the careful appraisal and constructive guidance. Below we respond point by point; all revisions are tracked in the manuscript.

Comments 1: 

The authors rely on anonymized self-reporting questionnaires. Hence, there is no chance to further investigate the potential correlation of dietary habits with imposed medical restrictions or administered treatment. This immediately implies that any attempt to explain the dietary patterns of hemodialysis patients (high fat, low carbohydrates, low fibers, high sodium, high phosphorus etc.) is highly elusive. The authors are kindly suggested to further clarify that issue on the ‘Limitations’ paragraph and reconsider revising their “Conclusion” section

Response 1: Thank you. We agree with this point. Therefore, we strengthened the Limitations and adjusted the Conclusion to remove any explanatory language and to reflect the cross-sectional, descriptive nature of our data.

Comments 2: 

The “Introduction” section is too long. The authors are kindly suggested to revise the text so as to be more concise.

Response 2: Agree. We shortened and refocused the Introduction to foreground the knowledge gap and study aim.

As part of these revisions, the entire manuscript underwent professional language editing by a native English speaker, with additional polishing after we shortened the Introduction and reframed the Conclusion.

Reviewer 3 Report

Comments and Suggestions for Authors

This is a very interesting and timely manuscript addressing dietary intake and nutritional imbalances in hemodialysis patients. The topic is highly relevant, particularly given the ongoing emphasis on individualized nutritional care in nephrology. 

However, to improve clarity, clinical applicability, and overall impact, several major and minor revisions are recommended.

Major Revisions

-The results section is highly detailed but difficult to navigate. Consider reorganizing it with a clearer flow (e.g., macronutrients, electrolytes, sensitivity analyses) and emphasizing the most clinically relevant findings in the main text while moving supplementary details to appendices or figures.

The lack of correlation between dietary intake and laboratory markers (e.g., serum phosphorus, albumin) limits clinical interpretation. If these data cannot be added, explicitly discuss this limitation and suggest that future studies address it.

The conclusions should be strengthened by including a concise, visually clear summary (e.g., a table or figure) outlining practical recommendations for clinicians and dietitians.

Minor Revisions

• Some sentences are overly long and complex; simplifying them would improve readability.

• Ensure consistent terminology (e.g., always use “hemodialysis [HD]” after first definition).

• Ensure all tables have self-explanatory titles and clearly defined units.

• Update citations to the most recent KDOQI guidelines.

• Confirm that all abbreviations (e.g., PEW, IDPN) are defined at first mention.

• The abstract is too dense; consider summarizing key findings.

Author Response

We thank the Reviewer for the thoughtful appraisal and practical suggestions. Below we respond point by point; all revisions are tracked in the manuscript.

Comments 1:

The results section is highly detailed but difficult to navigate. Consider reorganizing it with a clearer flow (e.g., macronutrients, electrolytes, sensitivity analyses) and emphasizing the most clinically relevant findings in the main text while moving supplementary details to appendices or figures.

Response 1. Thank you for this helpful suggestion. While we kept the substantive content unchanged, we made light-touch structural edits to improve navigation and clinical focus

Comments 2:

The lack of correlation between dietary intake and laboratory markers (e.g., serum phosphorus, albumin) limits clinical interpretation. If these data cannot be added, explicitly discuss this limitation and suggest that future studies address it.

Response 2: Agree. Because data collection relied on anonymized questionnaires, laboratory linkage was not feasible. We now state this explicitly and clarify the interpretive limits. We also outline a plan for future work to integrate EHR data (serum phosphorus, albumin, potassium, sodium), dialysate parameters, phosphate-binder dosing, and multi-day recalls to enable patient-level correlation analyses.

Comments 3:

The conclusions should be strengthened by including a concise, visually clear summary (e.g., a table or figure) outlining practical recommendations for clinicians and dietitians.

Response 3: Thank you for this suggestion. We agree with the intent. In this round we did not add a new table/figure. Two constraints guided this choice: the most legible formats we tested (radar charts/heatmaps) were not reader-friendly with our FFQ distributions and the journal’s length/figure limits left little room for an additional visual without removing core results. To partially address your point, we tightened the Conclusions to end with a two-sentence, clinic-oriented takeaway (energy/protein adequacy checks; sodium label literacy and home-cooking swaps; avoidance of phosphate additives with timing of binders; potassium-safer vegetables with portioning/leaching; triggers for dietitian review).

Comments 4:

Some sentences are overly long and complex; simplifying them would improve readability.

Response 4: Implemented. We shortened long sentences, removed redundancies, and tightened topic sentences across Abstract, Methods, Results, and Discussion. Importantly, the entire manuscript underwent professional language editing by a native English speaker after these revisions.

Comments 5:

Ensure consistent terminology (e.g., always use ‘hemodialysis [HD]’ after first definition).

Response 5: Implemented. We define hemodialysis (HD) at first mention and use HD thereafter; related terms are harmonized.

Comments 6:

Ensure all tables have self-explanatory titles and clearly defined units.

Response 6: Implemented. Each table now has: (1) a descriptive title indicating content and population, (2) clearly specified units (e.g., g/day, mg/1,000 kcal, % of patients), and (3) consistent footnotes for abbreviations and cutoffs.

Comments 7:

Update citations to the most recent KDOQI guidelines.

Response 7: Thank you for this suggestion. We verified scope and currency. The most recent dialysis-specific nutrition guideline remains KDOQI Clinical Practice Guideline for Nutrition in CKD: 2020 Update. By contrast, the KDIGO 2024 CKD Guideline explicitly notes that people receiving dialysis and kidney transplant recipients are not the focus; therefore, it should not be used as the primary authority for maintenance hemodialysis nutrition targets.

Comments 8:

Confirm that all abbreviations (e.g., PEW, IDPN) are defined at first mention.

Response 8: Implemented. We added first-mention definitions for PEW (protein-energy wasting), IDPN (intradialytic parenteral nutrition), and other terms (e.g., spKt/V, FFQ).

Comments 9:

The abstract is too dense; consider summarizing key findings.

Response 9: Implemented. The Abstract now uses a tighter structure, reports the most clinically relevant contrasts and removes second-order details.

Reviewer 4 Report

Comments and Suggestions for Authors

General comments

It would be appropriate to report what the patients knew about their diet. Were clear instructions given for each parameter before the study (calories, proteins, fats, carbohydrates, electrolytes, etc.), i.e. did the patients know what they were supposed to consume as hemodialyzed patients? This would indicate their degree of compliance, which is important as information.

It is also good to mention the composition of the dialysate that the patients used (e.g. potassium, sodium, calcium and magnesium) in methods

While the authors mention calcium and magnesium in the study and results, they do not mention them in the discussion.

  1. Page 6, paragraph 4, line 7-8: The mean age of participants was 52 ± 9 years (this is very interesting, mean age of the patients to be so low). Is there any explanation?
  2. Page 11, paragraph 3, lines 1-4: It should be emphasized that low-protein diets, although initially perceived as beneficial for controlling nitrogen metabolism, may result in adverse health consequences, such as increased metabolic acidosis, oxidative stress, and disturbances in electrolyte balance.

Comment: low protein diets do not make metabolic acidosis

  1. Since the patients participating in the study received a relatively smaller amount of proteins, it would be appropriate to report the average serum albumin levels of all the patients (Mean ±+ SD).
  2. Page 2-4: Introduction - It is too large and should be limited.
  3. Page 15: Conclusions - It is too large and should be limited.

Author Response

We thank the Reviewer for the thoughtful appraisal and practical suggestions. Below we respond point by point; all revisions are tracked in the manuscript (professionally edited by a native English speaker).

Comment 1: It would be appropriate to report what the patients knew about their diet. Were clear instructions given for each parameter before the study (calories, proteins, fats, carbohydrates, electrolytes, etc.), i.e. did the patients know what they were supposed to consume as hemodialyzed patients?

Response 1: Thank you. We have clarified the unit’s structured nutrition education pathway in the Methods.

Comment 2: It is also good to mention the composition of the dialysate that the patients used (e.g. potassium, sodium, calcium and magnesium) in methods.

Response 2: Implemented. We added the unit-standard dialysate prescription to Methods

Comment 3: While the authors mention calcium and magnesium in the study and results, they do not mention them in the discussion.

Response 3 : Addressed. The Discussion now includes a dedicated subsection: Calcium and Magnesium Intake- Clinical Considerations.

Comment 4: Page 6, paragraph 4, line 7–8: The mean age of participants was 52 ± 9 years (this is very interesting, mean age of the patients to be so low). Is there any explanation?

Response 4: Thank you for pointing this out. This was a typographical error. After re-checking the dataset, the correct mean age is 62 ± 9 years.

Comment 5: Page 11, paragraph 3… low-protein diets… may result in … increased metabolic acidosis… Comment: low protein diets do not make metabolic acidosis.

Response 5: We agree and have rephrased the paragraph to prevent misinterpretation. The revised text states explicitly that low-protein intake does not by itself induce metabolic acidosis in maintenance HD

Comment 6: Since the patients… received a relatively smaller amount of proteins, … report the average serum albumin levels (Mean ± SD).

Response 6: We cannot provide albumin because the FFQ was fully anonymous and not linkable to laboratory records by design. We have made this explicit and emphasized that the absence of lab linkage limits clinical correlation

Comment 7: Page 2–4: Introduction - It is too large and should be limited.

Response 7: Implemented. The Introduction was shortened and refocused on the knowledge gap and study aims; non-essential background was removed a

Comment 8: Page 15: Conclusions -It is too large and should be limited.

Response 8: Implemented. The Conclusions were tightened to a concise, practice-oriented summary

Round 2

Reviewer 3 Report

Comments and Suggestions for Authors

We thank the authors for considering and adding the changes suggested in our recommendations in order to improve the quality of the manuscript, for my part the manuscript could be published.

Author Response

Dear Reviewer,

Thank you for your thoughtful review and positive recommendation. We appreciate your time and constructive guidance, which helped us improve the manuscript.

With kind regards,
Łukasz Czyżewski